# Can Hydropower Still Be Considered a Clean Energy Source? Compelling Evidence from a Middle-Sized Hydropower Station in China

**Xuerong Li [1], Faliang Gui [2],\* and Qingpeng Li [3]**

[1] Jiangxi Engineering Research Centre of Water Engineering Safety and Resources Efficient Utilization, Nanchang Institute of Technology, Nanchang 330099, China

[2] Scientific Research Office, Nanchang Institute of Technology, Nanchang 330099, China

[3] State Grid Nanchang Electric Power Supply Company, Nanchang 330012, China

\* Correspondence: 1986992280@nit.edu.cn; Tel.: +86-791-88125948

**Abstract:** The development of clean energy is of great importance in alleviating both the energy crisis and environmental pollution resulting from rapid global economic growth. Hydroelectric generation is considered climate benign, as it neither requires fossil carbon to produce energy nor emits large amounts of greenhouse gases (GHG), unlike conventional energy generation techniques such as coal and oil power plants. However, dams and their associated reservoirs are not entirely GHG-neutral and their classification as a clean source of energy requires further investigation. This study evaluated the environmental impact of the Xiajiang hydropower station based on life cycle assessment (LCA) according to the 2006 Intergovernmental Panel on Climate Change (IPCC) guidelines, focusing specifically on GHG emissions after the submersion of the reservoir. Results reveal that although hydropower is not as clean as we thought, it is still an absolute "low emissions" power type in China. The amount of GHG emissions produced by this station is 3.72 million tons with an emissions coefficient of 32.63 g $CO_2$eq/kWh. This figure is lower than that of thermal power, thus implying that hydropower is still a clean energy resource in China. Our recommendations to further minimize the environmental impacts of this station are the optimization of relevant structural designs, the utilization of new and improved construction materials, and the extension of farmland lifting technology.

**Keywords:** hydropower; sustainable energy; climate change; greenhouse gases; clean; emissions

## 1. Introduction

Hydropower is a renewable and clean energy source that is playing a vital role in the adjustment of China's energy structure, the reduction of its GHG emissions, and the country's efforts to offset climate change. The development of hydropower is also of great importance in alleviating both the energy crisis and environmental pollution resulting from China's rapid economic growth in the 21st century. In general, the Chinese government has attached great importance to the issue of global climate change. During the 2015 United Nations Climate Change Conference in Paris, the Chinese government committed to reducing its carbon intensity by 40–45% before 2020 and pledged to support other developing countries' efforts to combat climate change. Considering the economic, technical, and environmental benefits of hydropower, the Chinese government gives high priority to its development [1].

There are a wide variety of hydropower stations around the world, designed for different functions, leading to uncertainty regarding the greenhouse effects of hydropower development [2,3]. Despite this, studies around the world demonstrate that, as hydropower generation requires both a dam and a reservoir, significant amounts of GHGs are emitted during their construction [4,5]. Scientists in Brazil for example, gathered data on reservoirs demonstrating that these GHG emissions of hydropower

can exceed those of thermal power [4]. In another relevant study, researchers at Washington State University found that the building of a reservoir had intensified GHG emissions and their effects. Furthermore, submerged hydropower plants cannot absorb carbon dioxide and their decomposition emits the potent greenhouse gas, methane. In light of this and other research, it is inevitable that the question should arise as to just how clean hydropower is, and whether it is possible to conclude that hydropower is not as entirely clean as one might suppose [5]. Is this the case in China? What should be done in order to improve the environmental cleanliness of hydropower? To support the sustainable development of hydropower in China, evaluating the environmental effects of hydropower stations in China is urgently needed.

Many important studies have been carried out with respect to its environmental effects. Life cycle assessment (LCA) has been widely used in evaluating the environmental footprint of various power generation systems [6–12]. However, due to the existence of different types, capacities, and locations of hydropower projects, the results of LCA with respect to environmental performance are site-specific and fluctuate within a certain range [6,13,14]. The environmental impact of the large number of hydropower stations in China has been evaluated with results indicating that the greatest environmental effects were caused primarily during the construction stage of these plants. Comparing these results with those of other countries revealed that the station in question had an environmental performance comparable to similar MW scale stations in Thailand and Japan but worse than that of stations in Switzerland [15]. Research for GHG emissions of three different types of small hydropower schemes in India indicated that GHG emissions were dependent on the head and capacity at smaller scales; these results were also found to be useful in predicting life cycle GHG emissions based on capacity, head, and type of these small hydropower schemes [16]. Hanafi and Riman assessed the life cycle of a mini hydropower plant in Simalungun, Indonesia, with results showing that both carcinogenic marine and freshwater aquatic eco-toxicity are the highest environmental effects generated from the construction of such hydropower plants [17]. Kadiyala et al. evaluated the life cycle GHGs emissions from different hydroelectricity generation systems [18]. Pascale et al. assessed the life cycle of a community hydropower system in rural Thailand; the results suggested that small-scale hydropower was more environmentally friendly than diesel generators and grid connection alternatives, considering mostly economic and social factors. However, it should be noted that the environmental impacts of the rural electrification alternatives were not addressed [19]. Jungbluth et al. conducted life cycle inventories of hydroelectric power generation in Europe, demonstrating that, despite the omission of the environmental impacts of station infrastructure in their study, marine eco-toxicity, human toxicity, and global warming were the highest impact effects related to the operation of hydropower plants [20].

Previous studies have carried out valuable research on the environmental impacts of hydropower stations, but several shortcomings still exist in terms of their extent and scope. For instance, most of the previously mentioned studies focused on small installed capacity hydropower stations as research subjects, while largely ignoring hydropower stations with greater capacities. Life cycle boundaries within these studies were also limited to a small number of stages despite the considerations of these generally encompassing the manufacturing, transportation, construction, operation, and disposal stages as well. In addition, most of these studies did not evaluate the GHG emissions from reservoirs despite research indicating that reservoirs are an important source of GHG emissions [21–23]. As such, limiting the consideration of these factors may overestimate the environmental cleanliness of hydropower. In order to adequately assess just how clean hydropower stations are in substance, analysis at all life cycle stages, as well the evaluation the GHG emissions from larger reservoirs are imperative. This study aims fill this gap by contributing valuable information on the environmental cleanliness of a hydropower station with a larger reservoir in China, with the aim of improving the environmental performance of hydropower in general. This paper uses Xiajiang hydropower station as a primary case study, which is a middle-sized hydropower plant with a large reservoir, commonly referred to as "the Three Gorges" of the Jiangxi province. During its different construction stages, environmental protection measures such as resource recycling, field lifting, and vegetation transplantation were

taken to reduce GHG emissions and mitigate its results. These measures, however, do not account for the environmental impact of the plant's life cycle; this is an entirely different subject worthy of further study.

The main purposes of this paper are first to evaluate the environmental cleanliness of this hydropower station based on LCA methods; second, to compare this environmental performance with that of others worldwide; and third, to provide recommendations in order to reduce the environmental impact of this plant and hydropower in general. To achieve this, the following section provides the materials and methods, including a description of the methodology, an introduction of the research background, an analysis of the system boundary and inventory, as well as data sources used within it. Sections 3 and 4 presents the results along with their related discussions. The final Section 5 provides the conclusions of this paper as well as important policy implications.

## 2. Materials and Methods

This section includes the following: the description of methods to calculate GHG emissions amount, the introduction of the research background, the analysis of system boundary and inventory, as well as the data sources used within it.

### 2.1. Methods to Calculate GHG Emissions Amount

This study evaluates the cleanliness of hydropower by calculating the amount of GHGs emissions of hydropower stations based on LCA methods [5,14,24]. LCA is an analytical methodology that provides an assessment of the environmental impact of relevant products and technologies from a "cradle to grave" system perspective, utilizing the detailed input and output parameters that operate within the designated system boundaries [24]. A GHG emissions estimation model was built in order to evaluate the environmental cleanliness of the Xiajiang hydropower station, according to the available data of engineering quantities, material use, and energy consumption at each stage. The IPCC (Geneva, Switzerland) measured and evaluated the global warming effects caused by GHG emissions using the Global Warming Potential (GWP) (Kyoto, Japan). The GWP is based on $CO_2$, therefore as other GHG are measured, and they are then converted into the $CO_2$ equivalent according to the corresponding equivalent coefficient. In this study, the equivalent coefficients of $CO_2$, $CH_4$, and $N_2O$ are 1, 21, and 310 respectively [24]. Taking this into account, the total amount of GHG emissions can be calculated using Equation (1).

$$W = \sum_j \sum_i w_{ij} \times k_i \qquad (1)$$

In Equation (1), $w_{ij}$ is the amount $i$ of any kind GHG in stage $j$, and $k_i$ is the equivalent coefficient of greenhouse gas $i$.

In order to facilitate the comparison between different hydropower stations, the GHG emissions coefficient $E_S$ is used as an evaluation.

$$E_s = \sum_j w_j \times Q^{-1} \qquad (2)$$

In Equation (2), $E_S$ is the GHG emissions coefficient of a specific hydropower station during the entire life cycle with units g/kWh and $Q$ being the total amount of power generation with units kWh [25]. Generally speaking, the smaller the coefficient, the cleaner the hydropower station.

### 2.2. Background

China has some of the richest hydro resources in the world with a total theoretical hydropower potential of 694 GW [26]. Over the last six decades, China's hydropower has developed rapidly with some large hydropower stations already operating while they were still under construction. These include the Three Gorges Project (TGP) and Pumped-Storage Power stations [1]. As can be seen in

Table 1, by the end of 2018, the installed generation capacity of hydropower was 352.26 GW, ranking first in the world in terms of capacity, and accounting for 18.54% of China's total installed power generation capacity. The annual power generation exceeded 1.23 trillion kWh, reducing GHGs by about 880 million tons each year and equivalent to saving about 512.5 million tons of standard coal.

**Table 1.** The total installed generation capacity of China in 2018 Unit: GW.

| Power Generation Technologies | Thermal Power | Hydropower | Wind Power | Solar Power | Nuclear Power | Total Installed Capacity |
|---|---|---|---|---|---|---|
| Installed capacity | 1143.67 | 352.26 | 184.26 | 174.63 | 44.66 | 1899.67 |
| Proportion | 60.21% | 18.54% | 9.70% | 9.20% | 2.35% | 100.00% |

Data source: Operation of China's Electric Power Industry in 2018.

The Xiajiang hydropower station (also called Xiajiang hydro-junction project), founded in September 2009, is located in the middle reaches of the Ganjiang River (as can be seen in Figure 1) in Ji'an City in the middle of Jiangxi province in China. It was designed for different functions, namely flood control, power generation, shipping facilitation, and irrigation. It has a reservoir capacity of 1.19 billion m$^3$, a normal reservoir area of 119 km$^2$, and an average water head of 46.00 m. The hydropower station, designed for a 100-year lifespan, is composed of nine bulb tubular turbine generators with a total installed generation capacity of 360 MW. In order to protect local farmland from being submerged by the reservoir, one of the farmland-lifting projects was implemented raising a total area of 903.13 ha. This project reduced the inundated area of cultivated land and forestland by 2519.4 ha and 245.2 ha respectively, protecting the sustainable utilization of land resources and reducing GHG emissions.

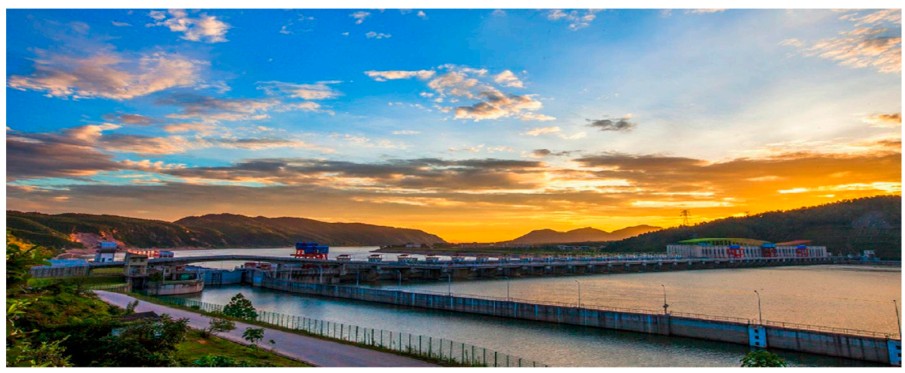

**Figure 1.** The Xiajiang hydropower station.

The first power unit was equipped with power generation capabilities in July 2013, with of the total nine power units entering full power production by March 2015. The main project was completed by August of the same year. The amount of engineering materials used throughout the construction of the entire project is listed in Table 2.

**Table 2.** The consumption amount of main engineering materials Unit: m$^3$, ton.

| Engineering Materials | Quantity |
|---|---|
| Earth-Rock Excavation | $3.90 \times 10^6$ m$^3$ |
| Earthwork Landfill | $2.67 \times 10^6$ m$^3$ |
| Concrete Casting | $1.19 \times 10^6$ m$^3$ |
| Metal Structure Installation | $3.87 \times 10^4$ m$^3$ |
| Cement | $3.27 \times 10^5$ t |
| Steel | $4.31 \times 10^4$ t |
| Diesel Fuel | $1.32 \times 10^4$ t |
| Explosive | $1.20 \times 10^3$ t |

Data source: Design report. Note: "$3.90 \times 10^6$ m$^3$" equals to 3,900,000 cubic meters, "$3.27 \times 10^5$ t" equals to 327,000 tons.

## 2.3. System Boundaries

The life cycle of this hydropower station consists of five distinct stages: the materials and equipment manufacturing stage, the transportation stage, the construction and equipment installation stage, the operation and maintenance stage, and the disposal stage. As shown in Figure 2, each stage requires the input of different materials, uses different energy sources, and releases GHGs at the same time. It must be pointed out that although this station is also designed for other functions such as flood control and irrigation, the GHG emissions from these other functions were not taken into account.

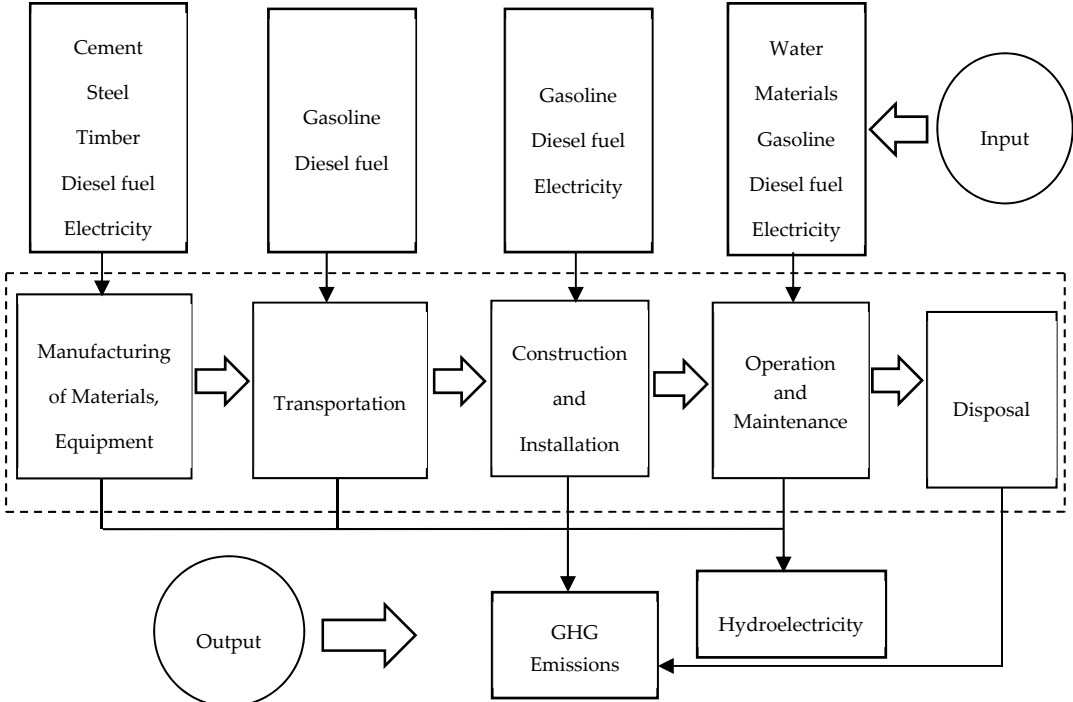

**Figure 2.** System boundary of this hydropower station.

## 2.4. Inventory Analysis

### 2.4.1. Manufacturing of Materials and Equipment

During the construction process of any hydropower plant, energy use, raw construction materials, electromechanical equipment, and the manufacturing of metal structural components all release GHGs at some point during the processes they are involved in respectively. As such, many developed countries have more sophisticated and detailed databases of standardized inventory for the environmental impact of construction materials. Due to the strong regionalism and timeliness of inventory data, many domestic scholars have conducted research on the environmental load of the most commonly used construction materials; however, a widely recognized inventory list or database has not yet been developed. Using the existing research results in China [27,28], this study considered the emissions data of the main energy and construction materials used during the manufacturing process of this station. For further details see Tables 3 and 4.

**Table 3.** Greenhouse gases (GHG) emissions list of major energy Unit: ton.

| Process | Unit Product | Main GHG Emissions | | |
|---|---|---|---|---|
| | | $CO_2$ | $CH_4$ | $N_2O$ |
| Produce | 1 kWh electricity | $8.92 \times 10^{-4}$ | $6.52 \times 10^{-9}$ | $8.82 \times 10^{-9}$ |
| | 1 kg diesel fuel | $6.53 \times 10^{-4}$ | $1.24 \times 10^{-8}$ | $6.41 \times 10^{-9}$ |
| Use | 1 kWh electricity | $6.24 \times 10^{-6}$ | $4.56 \times 10^{-10}$ | $6.17 \times 10^{-10}$ |
| | 1 kg diesel fuel | $3.10 \times 10^{-3}$ | $1.18 \times 10^{-7}$ | — |
| Total | 1 kWh electricity | $9.54 \times 10^{-4}$ | $6.98 \times 10^{-9}$ | $9.44 \times 10^{-9}$ |
| | 1 kg diesel fuel | $3.75 \times 10^{-3}$ | $1.30 \times 10^{-7}$ | $6.41 \times 10^{-9}$ |

Data source: 2006 IPCC national greenhouse gas emission list guides.

**Table 4.** GHG emissions list of major construction materials Unit: ton.

| Unit Materials | Main GHG Emissions | | |
|---|---|---|---|
| | $CO_2$ | $CH_4$ | $N_2O$ |
| 1 t Lime | $4.58 \times 10^{-4}$ | $2.01 \times 10^{-3}$ | $3.17 \times 10^{-6}$ |
| 1 t Cement | $5.74 \times 10^{-1}$ | $2.10 \times 10^{-3}$ | $3.30 \times 10^{-6}$ |
| 1 m$^3$ Natural Aggregate Concrete | $3.52 \times 10^{-1}$ | $4.60 \times 10^{-4}$ | $7.23 \times 10^{-7}$ |
| 1 m$^3$ Normal concrete block | $1.46 \times 10^{-5}$ | $4.28 \times 10^{-4}$ | $7.57 \times 10^{-7}$ |
| 1 m$^3$ Steel | $8.20$ | $1.80 \times 10^{-4}$ | — |

Data source: 2006 IPCC national greenhouse gas emission list guides.

Due to a lack of data, this study did not calculate the GHG emissions of the used mechanical and electrical equipment, focusing rather on the construction materials-based emissions.

### 2.4.2. Transportation

The transportation stage involves the movement of energy, construction materials, and other equipment from their respective manufacturers or distributors to the hydropower station via railway and highways. Due to the scale of this station and the logistic complexity of the transportation process, the proportion of each mode of transportation to the total transportation volume cannot be accurately determined. As diesel fuel was the primary energy source during this stage, its consumption amount was used to calculate the total GHG emissions in the transportation and construction stages. According to the design report, the total diesel consumption was 13,200 tons, and the GHG emissions coefficient of diesel during transportation and construction stage can be calculated by referencing the 2006 IPCC report. For details, see Table 5.

**Table 5.** The GHG emissions coefficient of diesel Unit: ton.

| | GHG Emissions | | |
|---|---|---|---|
| | $CO_2$ | $CH_4$ | $N_2O$ |
| Diesel emissions coefficient | $3.16$ | $1.28 \times 10^{-4}$ | $2.56 \times 10^{-5}$ |

Data source: 2006 IPCC national greenhouse gas emission list guides.

### 2.4.3. Construction and Installation

This stage encompasses infrastructure construction as well as electromechanical equipment and metal structure installation. These projects require a certain amount of labor and machinery, both of which generate a certain amount of GHG emissions. In this stage, energy consumption is also necessary, as both the construction and equipment installation processes require a certain amount of electrical power. As there is a variety of construction machinery with different energy requirements and units involved this stage, the total fuel consumption is represented by diesel oil. In addition,

because the life of the equipment was designed to be the same as that of the hydropower station, it was assumed that there were no large-scale equipment replacements during the entirety of the hydropower station life cycle, barring small repairs and maintenance. The energy consumption and environmental emissions of the energy production and use during the construction stage are listed in Table 3. This study assumed that the total amount of electricity used during this stage to be 60 million kWh.

### 2.4.4. Operation and Maintenance Stage

The GHG emissions during the operation and maintenance stages are separated into two distinct parts: the first is any emissions that resulted from the energy consumption required for the operation and maintenance of the power station, and the second is the emissions produced by the water body and biomass decomposition after the reservoir was impounded.

As for the first part, the hydropower station mainly consumed energy during the operation of auxiliary and power production equipment. The equipment mainly responsible for this consumption includes the turbine generator and related electrical equipment, a transformer, hydraulic hoists, water pumps, and other equipment needed for operating the station. Electricity consumption was required for lifting processes as well as office and equipment operation. GHG emissions in this stage were calculated by using operation power consumption figures as well as relevant energy coefficient equivalents, according to the supply ratio of hydropower and thermal power during the operation period. This study assumed the amount of diesel fuel and electricity consumed during the entire life cycle in this stage to be 50,000 tons and 500 million kWh.

As to the second part, an increasing number of studies show that reservoirs are significant sources of GHG emissions. When a reservoir is impounded, a large number of fields and plants are submerged, thereby releasing GHGs. In particular, the main sources of GHG emissions include natural reservoir discharges, spillways, and rivers downstream of the dam [29]. Reservoir GHG emissions are part of a complex process (see Figure 3) and relevant research results show that the changes of both GHG sources and sinks in different climate and geographical zones, follow strong case characteristics of reservoir operation areas [30,31]. Evidence from researchers suggests that the GHG emission from the decomposition of the organic matter resulting from the reservoir inundation occurs mainly during the first 10 years, reaches maximum levels at 2 to 3 years, and gradually decrease from then onward [29]. In addition, researchers have explored lake contributions to atmospheric GHGs by observing the GHG flux at the water–air interface in Poyang Lake. The results of these studies are of great significance to the calculation of GHG emissions in the study of hydropower reservoir emission, as they imply that the GHG emissions of the lake follow seasonal differences [32].

Considering that the mechanism of GHG emission from reservoirs is similar to that of lakes, and that the Xiajiang hydropower station belongs to the Poyang Lake River Basin sharing the same latitudes, this study draws on the research conclusions of lake GHG emissions to calculate the emission of this reservoir. As in the case of lake emissions, GHG emission from the reservoir inundation reaches a maximum at 2 to 3 years and demonstrating corresponding seasonal differences [3,4]. As such this paper assumes that the GHG emissions remain constant during the hundred-year lifespan, differing only by season. These seasonal GHG emissions are shown in Table 6.

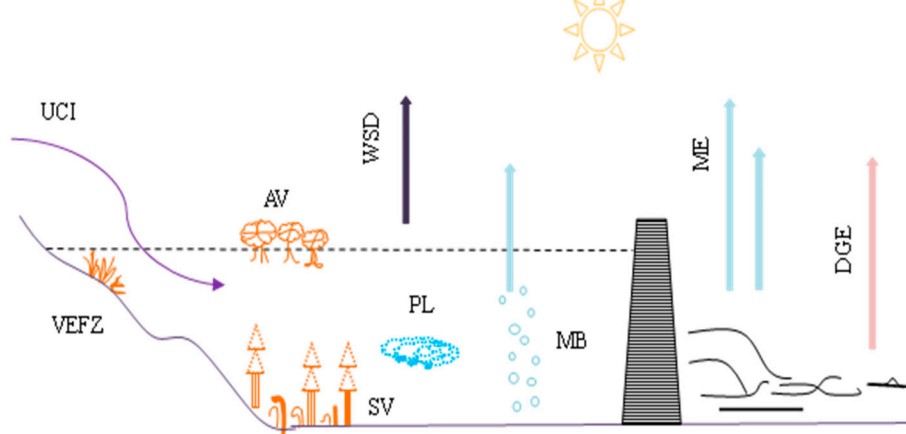

**Figure 3.** Emission approaches of GHG derived from the reservoir. Note: In Figure 3, UCI, VEFZ, AV, WSD, PL, SV, MB, ME, and DGE represent upstream carbon input, vegetation in the ebb and flow zone, aquatic vegetation, water surface diffusion, plankton, submerged vegetation, methane bubbles, methane escape, and downstream gas emissions, respectively.

**Table 6.** Seasonal GHG emissions of the reservoir.

| Season | GHG Emissions (g/m$^2$•d) | | |
|---|---|---|---|
| | $CO_2$ (A) | $CH_4$ (B) | $N_2O$ (C) |
| Spring ($Q_1$) | $3.97 \times 10^{-1}$ | $6.00 \times 10^{-3}$ | $6.85 \times 10^{-5}$ |
| Summer ($Q_2$) | $5.11 \times 10^{-1}$ | $1.23 \times 10^{-2}$ | $9.29 \times 10^{-5}$ |
| Autumn ($Q_3$) | $3.82 \times 10^{-1}$ | $6.96 \times 10^{-3}$ | $3.60 \times 10^{-5}$ |
| Winter ($Q_4$) | $1.75 \times 10^{-2}$ | $1.68 \times 10^{-3}$ | $5.40 \times 10^{-5}$ |
| Each day [$Q_d = (Q_1 + Q_2 + Q_3 + Q_4/)4$] | $3.28 \times 10^{-1}$ | $6.75 \times 10^{-3}$ | $6.30 \times 10^{-5}$ |

Data source: Lin, 2012. Greenhouse gas fluxes on the water–air interface of Poyang Lake. The inundation area of this reservoir is 119 million m$^2$. The total amount of GHG emission could be calculated using the above figures.

### 2.4.5. Disposal

In LCA studies, the disposal stage of a hydropower station is essential. As this hydropower station is unique in terms of its disposal stage, this study assumed that 30% of the equipment materials, totaling about 38,700 tons, were recycled and that the rest were sent to the nearby landfill for disposal. This assumption is based on separate cases of other renewable energy projects [33,34]. Approximately 2% of the initial weight of equipment that is in contact with water was assumed to be lost to water corrosion [33]. In addition, as most dams remain in their original construction site past their effective design lifespan, this study assumed that all of the building work would be left in place [35].

### *2.5. Data Sources*

In this study, the data included a primary and secondary data part. The primary data part was collected from the design report, and the secondary data part, i.e., the life cycle burden of the former input, such as equipment manufacturing, transportation, and electricity generation, was obtained from the China electric power-yearbook [36], the China energy statistical yearbook [37], 2006 IPCC guidelines [38], and other references [32,39]. It should be noted that most of the background data of the important inputs applied in this study, including cement, steel, diesel fuel, and electricity, were localized to China-specific parameters. This may cause some uncertainties in the evaluation results if tested elsewhere.

### 3. Results

The total GHG emissions of the engineering material utilized throughout the entire life cycle of the Xiajiang hydropower station was calculated to be 3.72 million tons, with an 80.11% concentration

of carbon dioxide. At present, the annual average power generation of this station is 1.14 billion kWh, and the annual average emissions of GHG are estimated to be about 0.77 million tons. As such, the emissions of GHG over the entire life cycle will theoretically reach 77 million tons; 3.72 million tons would be but a fraction of this amount. This demonstrates that even though hydropower is not a "zero emissions" power source, it is most definitely a "low emissions" energy type and is, therefore, still an option for clean energy in China. The emissions results for each stage are detailed in Table 7.

The GHG emissions coefficient was calculated to be 32.63 g $CO_2$eq/kWh, based on the previously mentioned data. In addition, in order to reduce the inundation area of the reservoir affecting the surrounding farmland, this project adopted filled lifting technology that reduced the inundation area to 3667.73 ha. The implementation of this technology has reduced the GHG emissions by 0.66 million tons.

**Table 7.** GHG emissions amount of Xiajiang hydropower station Unit: t.

| Stages | | GHG Emissions | | | Subtotal $CO_{2eq}$ $D = 1 + B*21 + C*310$ | Percentage |
|---|---|---|---|---|---|---|
| | | $CO_2$ (A) | $CH_4$ (B) | $N_2O$ (C) | | |
| Materials Manufacturing | | $9.77 \times 10^{+5}$ | $2.01 \times 10^{+3}$ | 2.10 | $1.02 \times 10^{+6}$ | 27.42% |
| Materials Transportation | | $4.17 \times 10^{+4}$ | 1.69 | $3.38 \times 10^{-1}$ | $4.19 \times 10^{+4}$ | 1.13% |
| Construction | | $1.33 \times 10^{+5}$ | 2.33 | $9.14 \times 10^{-1}$ | $1.33 \times 10^{+5}$ | 3.58% |
| Operation and Maintenance | Station Operation | $4.96 \times 10^{+5}$ | 4.14 | 4.75 | $4.96 \times 10^{+5}$ | 70.49% |
| | Reservoir Inundation | $1.43 \times 10^{+6}$ | $2.94 \times 10^{+4}$ | $2.74 \times 10^{+2}$ | $2.13 \times 10^{+6}$ | |
| Disposal | | $-9.33 \times 10^{+4}$ | $-2.05 \times 10^{+2}$ | — | $-9.76 \times 10^{+4}$ | −2.62% |
| Total | | $2.98 \times 10^{+6}$ | $3.12 \times 10^{+4}$ | $2.82 \times 10^{+2}$ | $3.72 \times 10^{+6}$ | 100.00% |

The GHG emissions percentages for each stage are shown in Figure 4. As can be seen in the figure, GHG emissions mainly occur in the operation and maintenance stage of the station's life cycle. The emissions during this stage account for 70.49% of the total life time emissions. The reservoir inundation period accounts for 81.11% of the emissions within this stage. In addition, the GHG emissions of the manufacturing stage comprise 27.42% of the total, as some materials used, such as cement and steel, release a large amount of GHG during this stage.

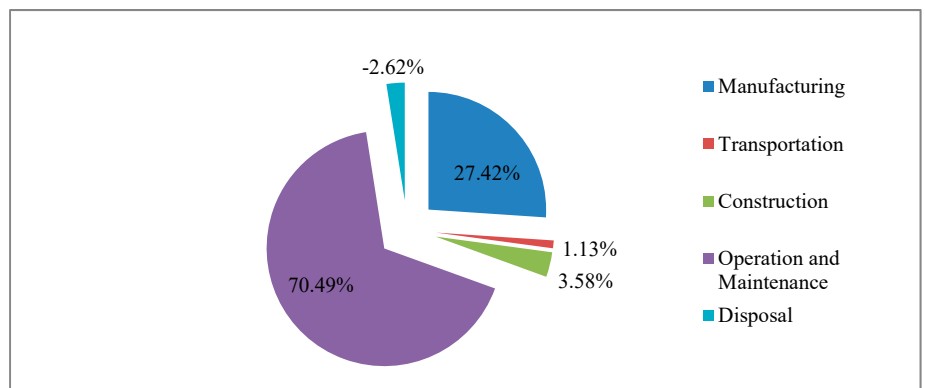

**Figure 4.** The proportion of GHG emissions at various stages of the Xiajiang hydropower station.

## 4. Discussion

### 4.1. Sensitivity Analysis

As cement, steel, electricity, and diesel fuel are the four dominating inputs in the entire lifecycle of this station, the environmental performance may vary with the consumption of the above four items; therefore, a sensitivity analysis is required. For each parameter, two scenarios were modeled and compared with the baseline: a 5% and 10% decrease of the total consumption to reduce environmental impact. As can be seen in Table 8, the variation of steel consumption influences the GHG impact most,

followed by the variation in cement consumption. A 10% decrease in steel consumption is estimated to reduce GHG emissions by 37,000 tons for a total steel contribution to GHG emissions' reduction of 1.060%.

**Table 8.** Sensitivity analysis results for Xiajiang hydropower station.

| Input | Amount (t, kWh) | Variation (%) | GHG Emissions | | | Sub-Total | Proportion |
|---|---|---|---|---|---|---|---|
| | | | $CO_2$ | $CH_4$ | $N_2O$ | | |
| Cement | $3.27 \times 10^{+5}$ | −5 | $-9.37 \times 10^{+3}$ | $-3.43 \times 10^{+1}$ | $-5.39 \times 10^{-2}$ | $-1.01 \times 10^{+4}$ | −0.275% |
| | | −10 | $-1.87 \times 10^{+4}$ | $-6.86 \times 10^{+1}$ | $-1.08 \times 10^{-1}$ | $-2.02 \times 10^{+4}$ | −0.530% |
| Steel | $4.31 \times 10^{+4}$ | −5 | $-1.77 \times 10^{+4}$ | $-3.88 \times 10^{+1}$ | — | $-1.85 \times 10^{+4}$ | −0.503% |
| | | −10 | $-3.53 \times 10^{+4}$ | $-7.76 \times 10^{+1}$ | — | $-3.70 \times 10^{+4}$ | −1.060% |
| Electricity | $6.00 \times 10^{+7}$ | −5 | $-2.85 \times 10^{+3}$ | $-4.40 \times 10^{-1}$ | $-5.90 \times 10^{-1}$ | $-3.04 \times 10^{+3}$ | −0.083% |
| | | −10 | $-5.70 \times 10^{+3}$ | $-8.80 \times 10^{-1}$ | −1.18 | $-6.08 \times 10^{+3}$ | −0.166% |
| Diesel fuel | $1.32 \times 10^{+4}$ | −5 | $-2.48 \times 10^{+3}$ | $-8.58 \times 10^{-2}$ | $-4.23 \times 10^{-3}$ | $-2.48 \times 10^{+3}$ | −0.068% |
| | | −10 | $-4.95 \times 10^{+3}$ | $-1.72 \times 10^{-1}$ | $-8.46 \times 10^{-3}$ | $-4.96 \times 10^{+3}$ | −0.136% |
| Total | — | −5 | $-3.24 \times 10^{+4}$ | $-7.36 \times 10^{+1}$ | $-6.48 \times 10^{-1}$ | $-3.41 \times 10^{+4}$ | −0.929% |
| | | −10 | $-6.47 \times 10^{+4}$ | $-1.47 \times 10^{+2}$ | −1.30 | $-6.82 \times 10^{+4}$ | −1.854% |

*4.2. Uncertainty*

The results of this study have some uncertainties. The first is that the GHG emissions list of energy and main construction materials is based on Chinese standards that may differ from international standards. The second is that the GHG emissions of some materials have not been calculated because of a lack of emissions inventories. The third is that the calculation of GHG emissions from the operation and maintenance stage is inaccurate because there are no accurate data of the energy and electricity being transported. Regarding these uncertainties, this study has made some assumptions to overcome the impact on the results.

## 5. Conclusions

This study evaluated the environmental impacts throughout the life cycle of the Xiajiang hydropower station based on life cycle assessment. Several concrete conclusions are drawn as follows:

The evaluation results confirm that hydropower station does indeed produce some negative environmental impacts. The life cycle environmental impact per kWh electricity produced by the station was evaluated to have a GWP of 32.63 g $CO_2$eq, with the operation and maintenance stage being the largest contributor of emissions, followed by materials manufacturing stage. Steel, cement, electricity, and diesel fuel are considered the four dominating inputs for the measured environmental impacts. Although this station performs worse than comparable stations in Turkey, comparing the results with those of thermal power generation technology shows that the emission coefficient of hydropower is much smaller; thus proving that, even if hydropower is not "zero emissions", it is well within the "low emissions" parameters for power types in China.

The results also demonstrate that hydropower is not as "clean" as we thought before. Therefore, the environmental impact must be reduced in the future, in order to improve the environmental performance of hydropower stations and promote the sustainable development of the hydropower industry in China.

Two important policy implications were concluded through this study. First, that the structural design of hydropower stations should be optimized to minimize the requirement of construction materials, in order to significantly reduce the GHG emissions. As revealed by the sensitivity analysis, a 10% decrease of steel consumption could decrease the amount of GHG emissions by 37,000 tons. Moreover, environmental-friendly construction materials can be applied to achieve further improved environmental performances, such as the substitution of cement for geo-polymer. This substitution could potentially reduce GHG emissions in the process of cement production. Second, measures should be taken to minimize the inundation of farmland and the respective vegetation by the reservoir. Farmland lifting is a new technology that deals with the issue of farmland inundation in China.

This technology not only reduces the area of inundated farmland, but also protects the environment and allows for the sustainable development of this and other resources. Under the premise that comprehensive benefit is feasible, the applications of this technology can be widely popularized in the future.

While this study made significant advancements with respect to the environmental impact of the Xiajiang hydropower station, it still had its limitations. Among these are the system boundary not including the GHG emissions during machinery and equipment production process, the energy emissions list being limited to Chinese standards, and key assumptions about the disposal stage. As such, there still exists a gap between the results of this study and the actual data. Therefore, it is necessary for future investigations to take into full consideration as many of these factors as possible, striving to make the research results more accurate.

**Author Contributions:** Q.L. collected the data. Both X.L. and F.G. analyzed the results with economic interpretation. All authors read and approved the final manuscript.

**Funding:** The APC was funded by the key program for the Jiangxi Provincial Hydraulic Science and Technology Projects (KT201726).

**Acknowledgments:** We are grateful to the anonymous reviewers for their helpful comments on the manuscript. Meanwhile, the support of the Science and Technology Project Founded by the Education Department of Jiangxi Province (GJJ171018) and the key program for the Jiangxi Provincial Hydraulic Science and Technology Projects (KT201726) are gratefully acknowledged.

**Conflicts of Interest:** The authors declare no conflict of interest.

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
