# Peer review of "Can Hydropower Still Be Considered a Clean Energy Source? Compelling Evidence from a Middle-Sized Hydropower Station in China"

_sustainability, doi:10.3390/su11164261_

Round 1

Reviewer 1 Report

This paper, despite the limitations that the authors comment on in the text, in my opinion, is very well structured, with suggestive contents that seek to reflect on the capacity for improvement in the design, planning and construction of hydroelectric power plants.

Simply, I think the authors should make more references to mini-hydro power plants (Small Hydropower). Could these small hydroelectric plants be an alternative to sustainability?

The authors could briefly discuss this topic.

Perhaps, some references of interest could be the following:

World Small Hydropower Development Report 2016. United Nations Industrial Development Organization, Vienna; and International Center on Small Hydro Power, Hangzhou.

- Espejo Marín, C.; García Marín, R; Aparicio Guerrero, A.E. (2017): “El resurgimiento de la energía minihidráulica en España y su situación actual”. Revista de Geografía Norte Grande, nº 67, pp. 115-143. (http://www.scielo.cl/pdf/rgeong/n67/art07.pdf) [The resurgence of small hydro-power in Spain and its current situation]

Author Response

We would like to thank the reviewer for the helpful comments on our manuscript. Our responses together with the reviewer’s comments are given below point by point.

    Simply, I think the authors should make more references to mini-hydro power plants (Small Hydropower)?

The authors could briefly discuss this topic.

Perhaps, some references of interest could be the following:

⇒Response:

Thank you for this useful comment. We have read and quoted some valuable references, and compared our findings with those studies.

Although the development of mini-hydro power plants can produce clean energy, which is conducive to the sustainable development of human society, it must be pointed out that mini- hydro power plants also cause some damage to the ecological environment, so we think these small hydroelectric plants cannot be an alternative to sustainability. Therefore, in the process of development and utilization, it is necessary to minimize the damage of mini-hydropower to the ecological environment, and explore other sustainable development programs.

The following references have been quoted.

[2] Espejo Marín, C.; García Marín, R; Aparicio Guerrero, A.E. (2017): “El resurgimiento de la energía minihidráulica en España y su situación actual”. Revista de Geografía Norte Grande, nº 67, pp. 115-143.

[41] World Small Hydropower Development Report 2016.

Reviewer 2 Report

The authors present an interesting paper about the hydropower energy source and about the possibility to consider it as a clean energy source. Specifically, the paper illustrates some aspects related to a middle-sized hydropower station in China.

There are major issues that should be corrected.

The manuscript presented by the authors does not have the structure and the style of Sustainability Journal. Authors should follow the specific section in the Journal’s website, entitled “Instructions for Authors”, with clear indications for the authors in relation to the structure and content of the paper that must be organized according to the following chapters: Introduction, Materials and Methods, Results, Discussion, Conclusions, Patents. So the referee suggests to reorganize the whole paper from the point of view of the structure and of the contents. In particular, the aspects of the research performed by the authors, the recognizable aspects of originality must be highlighted, in addition to the fact that the criteria for replicability of the illustrated method must be clearly explained.

Other main areas to be improved are given below:

-       The abstract needs a minor revision. It is necessary to incorporate all the main components of the manuscript: background, methods, results and conclusions.

-       In the introduction, especially at the beginning, many sentences are not supported by references (for example “studies around the world demonstrate…”). Provide reference for these sentences.

-       The introduction and the background must be summarized according to the style and format required by the journal. Furthermore, other significant references must be added.

-       It must be better explained what your research questions are. Moreover, research gap is not clear from introduction section. The introduction section also needs revision in order to rationalize that why this study should be conducted. Focus on explaining why the current study is important, and its significant contributions in literature.

-       The methodology must precede the case study background. At the same time all the hypotheses made and the calculation references used must be contained in the methodology.

-       In the table 2 explain the unit of measurement. The same for the others tables.

-       Methodology should be support by relevant literature.

-       The data reference in the chapter 4.2 should be cited in its context.

-       The caption of figure n. 2 should better describe the figure itself.

-       In the methodology and results sections, it is suggested to insert an explanation of the structure of the chapter in sub-chapters.

-       The critical discussion is missing. 

Based on the following highlighted areas, the manuscript needs major revisions. Once these revisions are done, it should be reviewed again to ensure its quality.

Reviewer 3 Report

Referee report on manuscript "Can Hydropower still be Considered a Clean 2 Energy Source? —Compelling evidence from a 3 Middle-sized Hydropower Station in China” by Xuerong Li et al.

The manuscript describes a method for calculating greenhouse gas emissions of a hydropower plant. The focus lays on the situation in China and considers the full lifecycle of the power station. The manuscript is of interest due to its contribution to the discussion on the low-emission power stations. The method bases on several assumptions but the uncertainties are discussed. The manuscript is well written and organized and can be accepted for publications.

I have only a few optional remarks:

-lines 55-56: Please state why do you chose China and Xiajang. What is the special situation there or, in contrast, why this is representative for other power stations.

-reference 23: Please add an URL

-Table 2. Please specify the data source “Design report” by indicating a citation and an URL in the bibliography

-Please explain shortly how “filled lifting technology” or farmland lifting technology works

-lines 364-365: There is a misfit between the reduction of 54.25% and the information given in Table 9

Author Response

Responses to Reviewer #3’s Comments

We would like to thank the reviewer for the helpful comments on our paper. Our responses together with the reviewer’s comments are given below point by point.

-lines 55-56: Please state why do you chose China and Xiajang. What is the special situation there or, in contrast, why this is representative for other power stations.

Response: China has some of the richest hydro resources in the world, by the end of 2018 the installed generation capacity of hydropower was 352.26 GW, ranking first in the world. The Xiajiang station is a middle-sized hydropower plant with a large reservoir, commonly referred to as “the Three Gorges” of the Jiangxi province. During its different construction stages, unlike other hydropower stations, some environmental protection measures such as resource recycling, field lifting, and vegetation transplantation were taken to reduce GHG emissions and mitigate its results.

-reference 23: Please add an URL

Response:  URL: http://news.bjx.com.cn/html/20180207/879489.shtml

-Table 2. Please specify the data source “Design report” by indicating a citation and an URL in the bibliography

Response:  We have add a citation in the bibliography, which is “[39] Feasibility Study Report of Xiajiang Water Conservancy Project in Jiangxi Province”, and this report is not yet available online, so we cannot give an URL.

-Please explain shortly how “filled lifting technology” or farmland lifting technology works

Response: “Filled lifting technology” is a new technology, its purpose is to protect cultivated land resources by raising part of farmland from being submerged while constructing the dam, the land can also grow crops, and this is a sustainable development measure.

-lines 364-365: There is a misfit between the reduction of 54.25% and the information given in Table 9

Response:  Thank you for pointing out this serious mistake. And we have changed 54.25% to 1.060%, that is “A 10% decrease in steel consumption is estimated to reduce GHG emissions by 37 thousand tons for a total steel contribution to GHG emissions reduction of 1.060%.”

You can also see the attachment, thank you!

Round 2

Reviewer 2 Report

Dear authors I really appreciate your revised paper. Related your suggestions request I think that the discussion is now correct but you should improved it removing from the results section all the comparisons with references (for example line 328 "Comparing...." about Table n.8, and so on).

Author Response

Dear authors I really appreciate your revised paper. Related your suggestions request I think that the discussion is now correct but you should improved it removing from the results section all the comparisons with references (for example line 328 "Comparing...." about Table n.8, and so on).

Response: Thank you for this helpful comment. Some of the contents are really irrelevant to this parts. We have revised them according to your suggestion. Thank you again for your valuable comments.